# Clinical outcomes of endoscopic submucosal dissection for colorectal neoplasms: A single-center experience in Southern Taiwan

Chen-Yu Ko[1], Chih-Chien Yao[1,2], Yu-Chi Li[1], Lung-Sheng Lu[1], Yeh-Pin Chou[1], Ming-Luen Hu[1,2], Yi-Chun Chiu[1,2], Seng-Kee Chuah[1,2], Wei-Chen Tai[1,2]*

1 Division of Hepato-Gastroenterology, Department of Internal Medicine, Chang Gung Memorial Hospital, Kaohsiung, Taiwan, 2 Chang Gung University College of Medicine, Kaohsiung, Taiwan

* luketai1019@gmail.com

## Abstract

### Background and aims

Endoscopic submucosal dissection (ESD) as an advanced endoscopic procedure can be considered for the removal of colorectal lesions with high suspicion of limited submucosal invasion or cannot be optimally removed by snare-based techniques. We aimed to analyze the clinical outcomes of ESD for colorectal neoplasms in our hospital.

### Methods

We retrospectively enrolled 230 patients with 244 colorectal neoplasms who received ESD procedures from April 2012 to October 2020 at Kaohsiung Chang Gung Memorial Hospital. Clinicopathological data were collected by chart review. We also recorded ESD-related complications and clinical outcomes.

### Results

The average age was 64 years old, with a mean follow-up time of 22.59 months. There was a loss of follow-up in 34 lesions. Most lesions were lateral spreading tumors of the non-granular type. The average ESD time was 51.9 minutes. Nine cases (3.7%) had procedure-related complications, including two intra-procedure perforations (0.8%) and seven delayed bleeding (2.9%) without procedure-related mortality. 241 lesions (98.8%) achieved en-bloc resection, while 207 lesions (84.8%) achieved R0 resection. Most lesions were tubulo-(villous) adenoma. Malignancy included 35 adenocarcinomas and 5 neuroendocrine tumors. No local recurrence was developed during follow-up. Multivariate analysis for long ESD time revealed significance in size $\geq$ 10 cm$^2$ and endoscopist's experience < 3 years. Pre-ESD endoscopic ultrasound revealed good prediction in discrimination of mucosal (sensitivity: 0.90) and submucosal lesion (specificity: 0.67).

**Data Availability Statement:** All relevant data are within the paper.

 1 / 13

**Funding:** The authors received no specific funding for this work.

**Competing interests:** The authors have declared that no competing interests exist.

## Conclusions

ESD for colorectal neoplasms is an effective and safe technique. Size $\geq$ 10 cm$^2$ and endoscopist's experience < 3 years were significantly associated with long procedure time. Pre-ESD EUS provided a good prediction for colorectal neoplasms in invasion depth.

## Introduction

Adenomatous polyps are recognized as precursor lesions leading to the development of colorectal cancer (CRC). Complete colonoscopic removal of these polyps could prevent the occurrence and death from CRC [1, 2]. A well-known association exists between adenoma detection rates and interval CRC risk [3, 4]. However, local recurrence of colorectal tumors is a major problem after endoscopic resections. The risk factors for local recurrence are tumor size, depth of tumor invasion, high-grade dysplasia polyps, piecemeal resection, villous tumor components, and positive histopathological margin [5–7]. Nowadays, there are various endoscopic techniques for the resections of these premalignant colorectal neoplasms, such as polypectomy, endoscopic mucosal resection (EMR), and endoscopic submucosal dissection (ESD) [8]. EMR was developed as a less invasive endoscopic option for removing lesions that cannot be snared by conventional methods, such as sessile or flat neoplasms confined to the superficial layers [9]. However, EMR is not an ideal technique for larger lesions due to the higher possibility of piecemeal resection and local recurrence [10].

Compared to EMR, ESD allows a better rate of en-bloc resection and reduces the local recurrence rate [11]. However, ESD is technically demanding, time-consuming, cost-intensive, and has a higher procedure-related complication rate [12, 13]. Despite the advantages of ESD, western countries infrequently chose ESD over EMR based on the greater technical difficulty involved, longer procedure times, and increased risk of perforation [14]. While diagnosing invasion depth, the macroscopic type and growth type of the lesion influence the accuracy rate of deep submucosal invasion [8]. Thus, the appropriate diagnostic methods, like endoscopic observation and endoscopic ultrasonography (EUS), are quite important [8].

For the usefulness of ESD and the role of EUS in diagnosis, we aimed to analyze the clinical outcomes of ESD for colorectal neoplasms and the accuracy of pre-ESD EUS in our hospital retrospectively.

## Materials and methods

We retrospectively reviewed 230 patients with 244 colorectal neoplasms who received colorectal ESD from April 2012 to October 2020 at Kaohsiung Chang Gung Memorial Hospital. None of these patients received previous EMR or polypectomy. Sixteen cases underwent biopsy only, which were unrelated to submucosal fibrosis during ESD. These colorectal lesions selected for ESD had morphological features, such as large broad-base or flat polyps, lateral spreading tumors, and submucosal tumors when EMR may result in piecemeal resection in advance. We chose ESD over EMR in some colorectal lateral spreading tumors less than 20 mm in cases of suspected lesions with limited submucosal invasion or difficult locations for en-bloc EMR, such as ileocecal valve, hepatic/splenic flexure, and sigmoid colon. We also performed an image-enhanced colonoscopy with a narrow band image and indigo-carmine dye spray as assistance to determine the invasion depth via NICE (NBI International Colorectal Endoscopic) and JNET (Japan NBI Expert team) classification [15, 16]. Lesions suspected of

advanced submucosal invasion (NICE or JNET classification type 3) were excluded. The study protocol was approved by the Institutional Review Board of Kaohsiung Chang Gung Memorial Hospital (IRB No.:202200582B0). The need for written informed consent was waived due to its retrospective, single-center nature.

Baseline characteristics of patients analyzed were age, gender, size of lesions, the gross appearance of lesions, location of lesions, pre-ESD endoscopic ultrasound results, types of anesthesia, ESD-related complications, and mean follow-up months. We used maximal length multiplied by maximal width to represent the size of the lesions. Furthermore, we analyzed the time of endoscopic submucosal resection.

Although EUS is not a routine exam before colorectal ESD in our hospital, pre-ESD EUS was performed for lesions with bigger sizes or central depression resulting in poor observation of the surface and micro-vascular pattern. EUS procedures were performed by two experienced endoscopists who have performed more than 2000 EUS procedures. Our EUS procedures used a miniature Probe (UM-2R; Olympus Medical Systems, Tokyo, Japan) and an ultrasound system (EU-ME2 Premier Plus; Olympus Medical Systems, Tokyo, Japan).

The ESD procedures (Fig 1) were performed under general sedation or non-sedation by five experienced endoscopists. The equipment used included flexible endoscopes with a distal cap and the HybridKnife™ water-jet system (ERBE, Tubingen, Germany) or DualKnife-J™ electrosurgical knife (Olympus, Tokyo, Japan). Submucosal injection included normal saline with Bosmin and indigo-carmine in ESD with Hybridknife and Glycerol with Bosmin and indigo-carmine in ESD with DualKnife-J. We initially made a circumferential incision of the mucosal layer with an electrosurgical knife, followed by a dissection of the submucosa. In some cases, we used traction techniques to help with submucosal incisions. Hybrid-ESD was defined as resection completed using CaptivatorTM II Single Use Snare (Boston Scientific, Natick, MA, USA) after adequate submucosal dissection and circumferential incision. We performed direct coagulation for hemostasis with the electrosurgical knife or the Coagrasper™ Hemostatic Forceps (Olympus, Tokyo, Japan) during the procedure and after complete resection. Mucosal defects were closed with SureClip® (Micro-Tech, Nanjing, China). Sulcrafate gel was sprayed on the wound of ESD after adequate coagulation to observe the possible minor bleeder. Post-ESD specimens were sent for pathology and classified histologically based on WHO classification. R0 resection was defined as an en-bloc resection with histologically clear deep and peripheral margins.

For the efficiency of colorectal ESD, there was no consensus on the length of ESD procedure time. Most of our ESDs were done within 100 minutes, and we considered the experience in Japan [17]. Thus, we defined a long procedure time as more than 100 minutes.

## Results

### Patient characteristics and gross appearance of colorectal neoplasms

As shown in Table 1, 230 patients who underwent ESD for 244 lesions were included in this study (males: 67; mean age: 64.0 ± 9.1 years). The mean post-ESD observation period was 22.59 months, with a loss of follow-up in 34 lesions (13.9%). The mean tumor size in the 244 colorectal lesions was 7.83 ± 6.6 cm$^2$, and 23.8% (58/244) were larger than 10 cm$^2$. Regarding the tumor morphology, 89% were lateral spreading tumors (217/244; 9 were granular type, 150 were non-granular type, and 58 were mixed type), 9% (5/244) were polypoid lesions, and 2% (5/244) were submucosal tumors. The distribution of lesions was 67.2% (164/244), 23.4% (57/244), and 9.4% (23/244) at the right-side colon, left-side colon, and rectum, respectively.

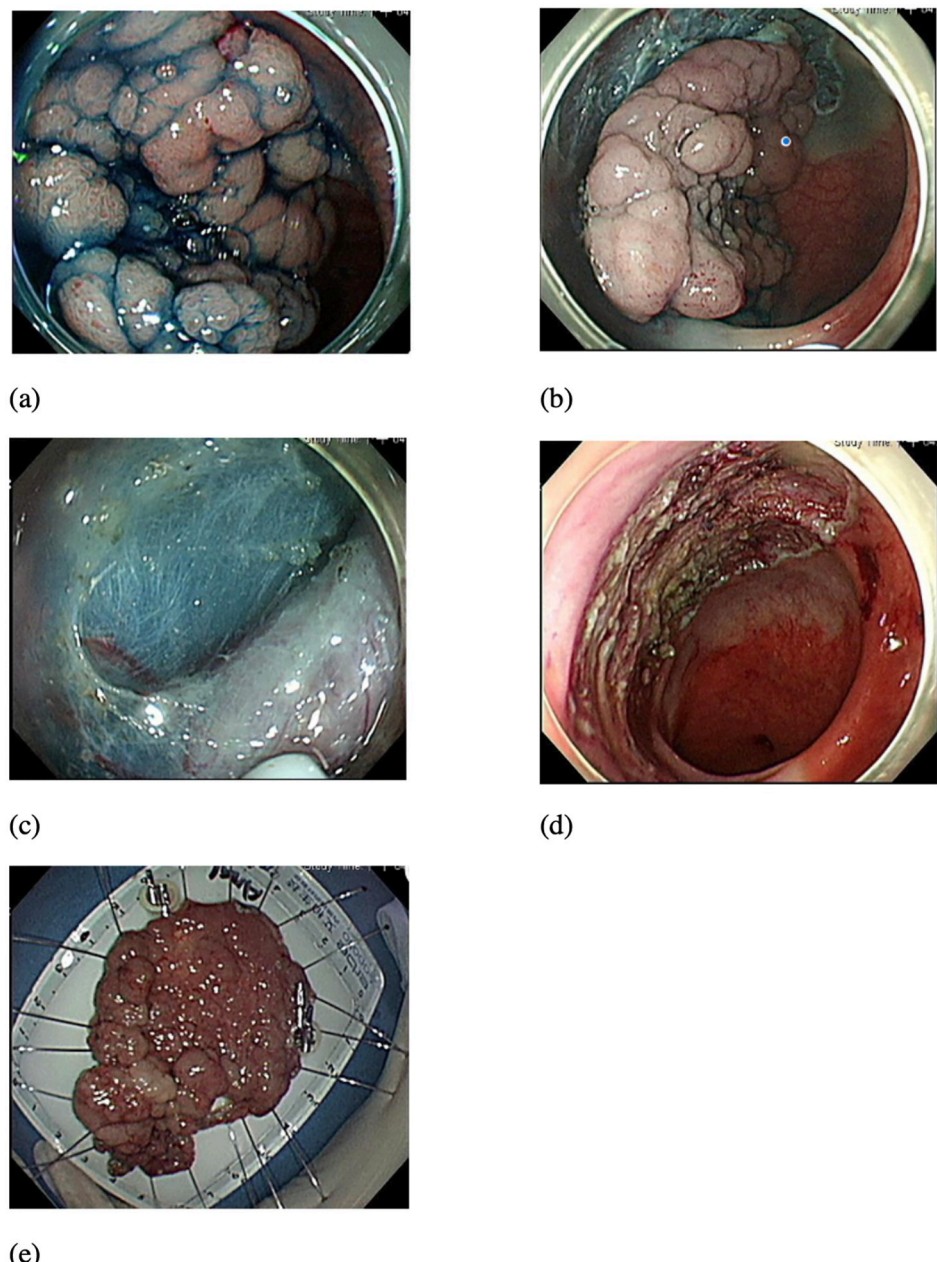

**Fig 1. Colorectal ESD procedure.** (a) Chromoendoscopy with indigo carmine staining (b) Circumferential cutting (c) Submucosal dissection (d) Complete resection (e) Specimen fixation.

### Therapeutic and histopathological results of ESD for colorectal neoplasms

The ESD time was 51.9 ± 32.9 minutes (Table 2). Forty of 244 lesions (16.4%) were completed with hybrid ESD. Nine cases (3.7%) had procedure-related complications, including two minimal perforations (0.8%) closed by SureClip® successfully after the complete ESD procedure and 7 delayed bleeding (2.9%). Our minimal perforation was defined as muscle layer defect without observation of mesenteric fat or intra-peritoneum organ and pneumoperitoneum. All post-ESD bleeding ceased spontaneously after medical treatment and observation. There was

**Table 1. Patient and gross appearance characteristics of colorectal neoplasms.**

| Total: 230 patients (244 lesions) | |
|---|---|
| **Patient characteristics** | |
| Age (years), mean ± SD | 64 ± 9.1 |
| Gender (male), n (%) | 67 (29.1%) |
| Mean follow-up times (months), mean ± SD | 22.59 ± 22.35 |
| Loss of follow-up, n (%) | 34(13.9%) |
| **Gross appearance of colorectal neoplasms** | |
| Size ($cm^2$), mean ± SD | 7.83 ± 6.6 |
| < 10 $cm^2$, n (%) | 186 (76.8%) |
| ≥ 10 $cm^2$, n (%) | 58 (23.8%) |
| Morphology | |
| Lateral spreading tumor (LST), n (%) | 217 (89%) |
| LST-G, Granular type, n (%) | 9 (3.7%) |
| LST-NG, Non-Granular type, n (%) | 150 (61.5%) |
| LST-MG, Mixed type, n (%) | 58 (23.8%) |
| Polypoid lesions, n (%) | 22 (9%) |
| Submucosal tumor, n (%) | 5 (2%) |
| Location | |
| Right colon, n (%) | 164 (67.2%) |
| Left colon, n (%) | 57 (23.4%) |
| Rectum, n (%) | 23 (9.4%) |

no procedure-related mortality. In this study, 241 lesions (98.8%) achieved en-bloc resection, while three cases (1.2%) converted to piecemeal EMR due to severe fibrosis. No local recurrence developed during follow-up (mean: 22.6 ± 22.35 months, minimum–maximum: 2–105 months), with loss of follow-up in 34 cases (13.9%).

The pathological features and results of 244 colorectal neoplasms are shown in Table 2, which comprised 204 (83.6%) premalignant neoplasms (159 were conventional polyps, and 45 were sessile serrated lesions) and 40 (16.4%) malignant neoplasms (35 were adenocarcinomas, and 5 were neuroendocrine tumors); 95.5% of tumor invasion depths were limited to the mucosal layer, while 10 lesions (4.1%) had submucosal invasion, and one (9.4%) had already invaded the muscle layer. Regarding the histopathological results, 207 lesions (84.8%) achieved R0 complete resection, and 37 lesions showed incomplete resection. Among the 37 cases without R0 resection, five (13.5%) were referred for further surgical interventions due to adenocarcinoma or high-grade dysplasia with submucosal invasion. One case of adenocarcinoma with submucosal invasion was lost to follow-up due to refusal of further surgical treatment. The remaining 31 cases included 26 cases of tubulo-(villous) adenoma and five intra-mucosal adenocarcinomas with mucosal margin involved by low-grade dysplasia. There was no local recurrence under endoscopic surveillance among these patients.

## The efficiency of ESD and the accuracy of pre-ESD EUS

We further analyzed the efficiency of ESD (Table 3). ESD time of the rectal lesions was significantly longer than left and right colonic lesions (80.2 ± 45.1, 43.7 ± 27.8, and 50.8 ± 30.5 minutes, respectively, p<0.001). Regarding size, ESD lesions bigger than 10 $cm^2$ had significantly longer ESD times (69.0 ± 35.8 and 46.6 ± 30.1, respectively, p <0.001). To analyze the influence of the endoscopist's experience, we divided the patients into three groups according to their years of experience: experience of < 3 years (2012–2014), experience of 3–5 years (2015–

**Table 2. Therapeutic and histopathological results of ESD for colorectal neoplasms.**

| Total: 230 patients (244 lesions) | |
|---|---|
| **ESD therapeutic results** | |
| ESD time (min), mean ± SD | 51.9 ± 32.9 |
| Hybrid-ESD, n (%) | 40 (16.4%) |
| ESD complication, n (%) | 9 (3.7%) |
| Perforation, minor, n (%) | 2 (0.8%) |
| Delayed bleeding, n (%) | 7 (2.9%) |
| En-bloc resection, n (%) | 241 (98.8%) |
| Overall local recurrence, n (%) | 0 (0%) |
| **Histopathological results** | |
| Premalignant neoplasm | 204 (83.6%) |
| Tubulo-(villous) adenoma | 159 (65.2%) |
| Sessile serrated lesions | 45 (18.4%) |
| HGD/LGD | 24/180 (9.8%/73.8%) |
| Malignant neoplasm | 40 (16.4%) |
| Adenocarcinoma | 35 (14.3%) |
| NET, grade 1 | 5 (2%) |
| Invasion depth | |
| Mucosal layer | 233 (95.5%) |
| Submucosa | 10 (4.1%) |
| Muscularis | 1 (0.4%) |
| R0 complete resection rate; n (%) | 207 (84.8%) |

ESD, endoscopic submucosal dissection; SD, standard deviation; HGD, high-grade dysplasia; LGD, low-grade dysplasia; NET, neuroendocrine tumor.

2017), and experience of 5–7 years (2018–2020). Decreasing ESD procedure time (p<0.001) along with an accumulation of ESD experience was significantly identified (Fig 2).

To detect the influence of factors on procedure time, we defined procedure time $\geq$ 100 min as a long ESD time. Univariate analysis showed that rectal lesions (odds ratio [OR]: 0.166; 95% confidence interval [CI]: 0.053–0.522, p = 0.002), lesion size $\geq$ 10 cm$^2$ ([OR]: 4.041; 95% CI:

**Table 3. Analysis of the efficiency of ESD.**

| Factors | ESD time (mean ± SD, min) | P-value |
|---|---|---|
| Location | | |
| Rectum | 80.2 ± 45.1 | <0.001 |
| Left-side colon | 43.7 ± 27.8 | |
| Right-side colon | 50.8 ± 30.5 | |
| Size | | |
| < 10 cm$^2$ | 46.6 ± 30.1 | <0.001 |
| $\geq$ 10 cm$^2$ | 69.0 ± 35.8 | |
| Endoscopist's experience | | |
| < 3 years | 81.4 ± 39.9 | <0.001 |
| 3–5 years | 55.7 ± 26.6 | |
| 5–7 years | 35.0 ± 22.4 | |

ESD, endoscopic submucosal dissection; SD, standard deviation.

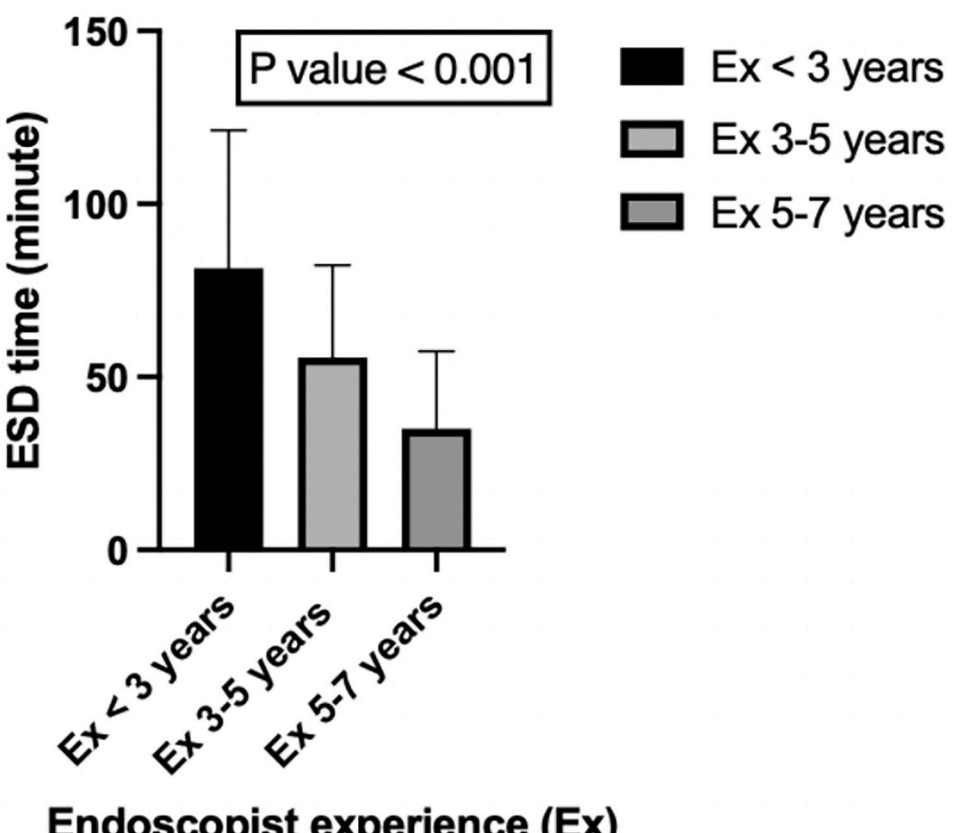

**Fig 2. Analysis of ESD time according to endoscopist's experience.**

1.481–11.024, p = 0.006), and endoscopist's experience < 3 years ([OR]: 0.032; 95% CI: 0.004–0.257, p = 0.001) were factors significantly associated with a long procedure time (Table 4)

Multivariate analysis was considered for all variables showing significant differences in the univariate analysis. Multivariate analysis for long ESD time revealed significant differences regarding lesion size ≥ 10 cm$^2$ ([OR]: 8.010; 95%CI: 2.097–30.591, p = 0.002), and endoscopist's experience < 3 years (odds ratio [OR]: 0.020; 95% CI: 0.002–0.201, p = 0.001) (Table 5).

Furthermore, we analyzed the accuracy of the pre-ESD EUS exam for lesions with indistinguishable invasion depth based on endoscopic appearance. Among 48 patients (19.7%) who received pre-ESD EUS, there were six cases with discordance (12.5%), including two lesions (4.2%) suspected as intra-mucosal carcinomas under EUS with histological submucosal invasion and four lesions (8.3%) suspected as focal submucosal invasion under EUS with histological intra-mucosal carcinoma. Despite the discordances, pre-ESD EUS showed good prediction in discriminating mucosal (sensitivity: 0.90, positive predictive value: 0.90) and submucosal lesions (specificity: 0.67, negative predictive value: 0.90) (Table 6).

## Discussion

### Therapeutic result of ESD for colorectal neoplasms

EMR was the minimally invasive, organ-sparing endoscopic technique considered to remove benign and early malignant colorectal lesions before ESD [9]. Several studies have compared ESD to EMR. ESD provides higher curative rates with better rates of en-bloc resection and R0

**Table 4. Univariate logistic regression analysis for long ESD time (procedure time ≥ 100 min).**

| Factors | N | OR | 95%CI | P-value |
|---|---|---|---|---|
| Age: < 60 years vs. ≥ 60 years | 244 | 1.804 | 0.501–6.490 | 0.367 |
| Gender: male vs. female | 244 | 1.680 | 0.530–5.325 | 0.378 |
| Procedure-related complication | 244 | 4.171 | 0.796–21.857 | 0.091 |
| Size: < 10 $cm^2$ vs. ≥ 10 $cm^2$ | 244 | 4.041 | 1.481–11.024 | 0.006 |
| Location | 244 | | | |
| Rectum | 23 | 0.166 | 0.053–0.522 | 0.002 |
| Left-side colon | 57 | 1.607 | 0.337–7.670 | 0.552 |
| Right-side colon | 164 | 1.000 | Ref. | |
| Morphology | 244 | | | |
| LST NG, non-granular type | 150 | 1.000 | Ref. | |
| LST MG, mixed type | 58 | 0.763 | 0.249–2.355 | 0.635 |
| LST-G, granular type | 9 | Not calculated | | |
| Submucosal tumor | 5 | Not calculated | | |
| Polypoid lesions | 22 | 1.511 | 0.184–12.414 | 0.701 |
| Histopathology | 244 | | | |
| Tubulo-(villous) adenoma | 159 | 1.000 | Ref. | |
| Sessile serrated polyps | 45 | 1.418 | 0.296–6.799 | 0.662 |
| Adenocarcinoma | 35 | 0.382 | 0.120–1.129 | 0.104 |
| NET, grade 1 | 5 | Not calculated | | |
| Invasion depth | 244 | | | |
| Mucosal layer | 233 | 1.000 | Ref. | |
| Submucosa | 10 | 0.593 | 0.070–5.037 | 0.632 |
| Muscularis | 1 | Not calculated | | |
| R0 resection | 244 | 1.935 | 0.593–6.318 | 0.274 |
| Endoscopist's experience | 244 | | | |
| < 3 years | 47 | 0.032 | 0.004–0.257 | 0.001 |
| 3–5 years | 94 | 0.173 | 0.020–1.505 | 0.112 |
| 5–7 years | 103 | 1.000 | Ref. | |

CI, confidence interval; ESD, endoscopic submucosal dissection; min, minute; LST, lateral spreading tumor; NET, neuroendocrine tumor; OR, odds ratio.

resection compared to EMR [11, 18]. However, due to the high technical demand for ESD, the outcome of ESD as en-bloc and R0 resection rates varied according to the hospital and country. A Korean institute revealed high en-bloc and R0 resection rates of 97.1% and 90.5%, respectively [19]. Another single-center research in European showed relatively lower en-bloc (88.4%) and R0 resection rates (62.6%) [18]. In this study, the ESD resection results were

**Table 5. Multivariate logistic regression analysis for the long ESD time (procedure time ≥ 100 min).**

| Factors | N | OR | 95%CI | P-value |
|---|---|---|---|---|
| Size: < 10 $cm^2$ vs. ≥ 10 $cm^2$ | 244 | 8.010 | 2.097–30.591 | 0.002 |
| Endoscopist's experience | 244 | | | |
| < 3 years | 47 | 0.020 | 0.002–0.201 | 0.001 |
| 3–5 years | 94 | 0.162 | 0.018–1.476 | 0.106 |
| 5–7 years | 103 | 1.000 | Ref. | |

CI, confidence interval; ESD, endoscopic submucosal dissection; min, minute. OR, odds ratio.

**Table 6. The accuracy of EUS.**

| | | Histologic diagnosis | |
|---|---|---|---|
| | | **Mucosal lesion** | **Submucosa lesion/invasion** |
| **EUS diagnosis** | **Mucosal lesion** | 38 | 2 |
| | **Submucosal lesion/invasion** | 4 | 4 |
| | | Sensitivity: 0.90 | Specificity: 0.67 |

EUS, endoscopic ultrasound.

98.8% en-bloc resection rate and 84.8% R0 resection rate without local recurrence. These results are consistent with that in a Northeast Asian country, illustrating ESD's effectiveness in colorectal neoplasms.

As for the result of colorectal ESD in the same area with a similar background, Choo et al. revealed en-bloc resection rate (72.7%) and R0 resection rate (66.7%) with perforation rate (15.2%) when the ESD technique was newly developed in Southern Taiwan [20]. A study by Tseng et al. showed en-bloc resection rate (90.2%) and R0 resection rate (89.1%) with perforation rate (12%) [21]. Although the studies mentioned above vary, improvements in en-bloc resection rate, R0 resection rate, and complication rate were observed in our study, contributing to the accumulation of ESD experience and improvement of ESD training programs in Southern Taiwanese hospitals.

The complete removal of colorectal adenomas reduces the risk of CRC [22], and the significant risk factor for local recurrence was a positive histopathological margin [5]. Two Japanese long-term studies revealed five-year local recurrence rates of approximately 1.5%, which was related to piecemeal resection and incomplete histologic resection [23, 24]. In our study, there was no local recurrence during follow-up, showing the consistency of a lower recurrence rate of ESD with other studies. On the other hand, immediate referrals to further surgical intervention for lesions without en-bloc resection or malignancy with unclear submucosal resection margin might contribute to promising results in our study.

When it comes to complications, the higher risk of ESD perforation is one reason EMR is more popular than ESD in western countries [14]. A meta-analysis showed higher complication rates for ESD than EMR (5.7% vs. 1.4%) [11]. The most common complications reported in another meta-analysis were bleeding (0.75%) and perforation (4.2%) [25]. Despite the higher risk of perforation with ESD, several studies stated that most perforations, either micro-perforation or macro-perforation, could be treated with endoscopic clipping without needing further surgical intervention, consistent with our study [26]. In our study, the total procedure-related complication was 3.7%, including delayed minor bleeding (2.9%) and minimal perforations (0.8%) without procedure-related mortality. Although there were higher rates of complications for ESD compared to conventional snare techniques, most complications were treated during ESD or conservative post-procedure care in this study.

## The efficiency of ESD

The longer procedure time compared to conventional snare-based techniques and EMR is another disadvantage of ESD [14]. We searched ESD outcomes from different countries as references. According to a single-center study in Japan, the mean ESD time was 46.4 minutes for 1199 lesions [17]. A study conducted by a Singapore group showed a mean time of 80.9 minutes with a median lesion size of 19.3 mm, showing that larger lesions require longer procedure times [27]. A study by a Korean group revealed a median time of 53.7 minutes for ESD, with a median lesion size of 21.1 mm [28]. A study in a western country had a median time of

105 minutes, with a median lesion size of 26 mm in 2011 [29]. Another recent study in a western country showed a median time of 82.7 minutes for ESD, with a median lesion diameter of 44.3 mm [30]. This study's average ESD procedure time was 51.9 minutes, with a median size of 7.83 cm$^2$, consistent with the study in a Northeast Asian country.

We aimed to detect the influential factors of long procedure times. Although a study in Japan revealed fibrosis as a factor influencing long procedure times [17], there was no statistical significance in our study due to the limited cases of submucosal fibrosis. We, therefore, analyzed the possible factors, including location, size, and endoscopist's experience. According to Gotoda et al., performing at least 30 cases is required to be an experienced endoscopist in ESD procedures [31]. We thus classified our ESD procedure into three groups based on the endoscopist's experience. Initial analysis revealed that rectal lesions, bigger lesion size $\geq 10$ cm$^2$, and endoscopist's experience were significantly related to long procedure times (Table 3). Univariate analysis showed significance in rectal lesions, lesion size $\geq 10$ cm$^2$, and endoscopist's experience $< 3$ years (Table 4). Multivariate analysis for long ESD time revealed significant differences in lesion size $\geq 10$ cm$^2$ and endoscopist's experience $< 3$ years (Table 5). Considering the ESD experience in Japan, lesion size was demonstrated as a factor related to longer procedure time, which is consistent with our results but not endoscopist's experience [17]. Miyaguchi et al. compared experienced experts to trainees, while we compared the difference in ESD experience in the same group of endoscopists [17]. In our study, gradually decreasing ESD time along with accumulating ESD experience in clinical practice was significantly identified (Fig 2).

### The accuracy of pre-ESD EUS

To evaluate the invasion depth of colorectal lesions, NICE and JNET classifications under chromoendoscopy with indigo carmine are widely used in clinical practice [15, 16]. The accuracy of deep submucosal invasion during ordinary or chromoendoscopic observation is around 70 to 80% [8]. According to current guidelines, EUS is not considered a routine examination before colorectal ESD [32]. A prospective study demonstrated that preoperative evaluation through EUS examination provided clues of possible pathological features and helped decide the treatment strategy [33]. The accuracy rate of EUS is approximately 80% in detecting deep submucosal invasion, which may help in diagnosis [8]. As diagnostic accuracy differs according to the macroscopic type and growth type of the lesion, appropriate diagnostic methods, such as endoscopic observation and EUS, should be combined depending on the situation [8]. In our experience, some circumstances lead to the poor observation of characteristics of these colorectal lesions, including lesions with bigger sizes or central depression. In these cases, the pre-ESD EUS results provided us with another aspect to determine the treatment plan. In our study, pre-ESD EUS revealed good prediction in discriminating mucosal (sensitivity: 0.90, positive predictive value: 0.90) and submucosal lesions (specificity: 0.67, negative predictive value: 0.90). The distortion of the lesions lead to the fuzzy boundary between the mucosa and submucosa, which might explain the discordances between EUS and histologic results.

### Limitations

Our study had several limitations. First, it was a single-center retrospective study conducted by five endoscopists. Second, only 85.3% of patients received surveillance colonoscopy. The rate of lost follow-up is 14.7%; therefore, some local recurrence may be undetected. Third, we did not analyze the different knives used in ESD. Fourth, various traction methods emerged and were demonstrated to be efficacious in facilitating ESD by maintaining satisfactory traction

during dissection [34, 35]. We also used traction methods in the colorectal ESD. However, due to the limited number of cases, we did not analyze the efficacy of these traction methods in the current study.

## Conclusions

Colorectal ESD is effective and relatively safe for colon mucosal lesions, lesions with possible superficial submucosal invasion, and lesions that snare-based techniques cannot optimally remove. This technique can allow high en-bloc resection rates and histologically R0 resection of large colorectal epithelial tumors and submucosal tumors with low complication rates. While considering the efficiency of ESD, lesion size $\geq 10$ cm$^2$ and endoscopist's experience were significantly associated with long procedure time. Pre-ESD EUS can provide a good prediction for colorectal neoplasms with uncertain NICE and JNET classification under endoscopic appearance and chromoendoscopy.

## Acknowledgments

The authors are grateful to all participating patients and their families.

## Author Contributions

**Conceptualization:** Chen-Yu Ko.

**Data curation:** Yu-Chi Li.

**Formal analysis:** Chen-Yu Ko.

**Investigation:** Chen-Yu Ko, Chih-Chien Yao, Wei-Chen Tai.

**Methodology:** Seng-Kee Chuah.

**Project administration:** Seng-Kee Chuah.

**Resources:** Wei-Chen Tai.

**Supervision:** Wei-Chen Tai.

**Validation:** Ming-Luen Hu, Yi-Chun Chiu.

**Visualization:** Lung-Sheng Lu, Yeh-Pin Chou.

**Writing – original draft:** Chen-Yu Ko.

**Writing – review & editing:** Chen-Yu Ko, Chih-Chien Yao, Wei-Chen Tai.

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
