## [Decision Letter · Decision Letter 0]

29 Aug 2022

PONE-D-22-23547Clinical outcomes of colon endoscopic submucosal dissection for colonic neoplasms: A single center experience in South TaiwanPLOS ONE

Dear Dr. Tai,

Thank you for submitting your manuscript to PLOS ONE. The manuscript was evaluated by two experts in this field and several questions raised. After careful consideration, we feel that it has merit but does not fully meet PLOS ONE’s publication criteria as it currently stands. Therefore, we invite you to submit a revised version of the manuscript that addresses the points raised during the review process.

Please include the following items when submitting your revised manuscript:A rebuttal letter that responds to each point raised by the academic editor and reviewer(s). You should upload this letter as a separate file labeled 'Response to Reviewers'.A marked-up copy of your manuscript that highlights changes made to the original version. You should upload this as a separate file labeled 'Revised Manuscript with Track Changes'.An unmarked version of your revised paper without tracked changes. You should upload this as a separate file labeled 'Manuscript'.

We look forward to receiving your revised manuscript.

Kind regards,

Hsu-Heng Yen

Academic Editor

PLOS ONE

Journal Requirements:

Additional Editor Comments:

  As the study aimed to study ESD outcome in Southern Taiwan, please provide short discussion for their result with other reports of colonic resection from the same area with similar background of ESD development, i.e. (Early report of ESD from Taiwan https://doi.org/10.1016/j.aidm.2015.01.002. ; Gastroenterol Res Pract. 2013;2013:891565. doi: 10.1155/2013/891565. Epub 2013 Feb 25.PMID: 23533391 )

Reviewers' comments:

Reviewer's Responses to Questions

**Comments to the Author**

1. Is the manuscript technically sound, and do the data support the conclusions?

Reviewer #1: Partly

Reviewer #2: Yes

2. Has the statistical analysis been performed appropriately and rigorously? 

Reviewer #1: Yes

Reviewer #2: Yes

3. Have the authors made all data underlying the findings in their manuscript fully available?

Reviewer #1: Yes

Reviewer #2: Yes

4. Is the manuscript presented in an intelligible fashion and written in standard English?

Reviewer #1: Yes

Reviewer #2: Yes

5. Review Comments to the Author

Reviewer #1: this article clearly showed authors experience about ESD. they concluded that EUS has clinical impact before ESD. However they did not mention the method of EUS (miniprobe or radial /linear scope). becaysee more than half of the pts lesion localized in right colon.

second point, thay should clearly mention that how many cases had endoscopic procedure bedore ESD (emr or tried for polypectomy in another center)

Reviewer #2: The aim of this retrospective study was to evaluate the efficacy and safety of ESD for colonic neoplasms. Some revisions are suggested before consideration for publication.

1. “Clinical outcomes of colon endoscopic submucosal dissection for colonic neoplasms: A single center experience in South Taiwan” => Revise as “Clinical outcomes of endoscopic submucosal dissection for colorectal neoplasms: A single center experience in Southern Taiwan.”

2. Please shorten the background and study aim in the abstract part.

3. Please revise “colonic” neoplasm as “colorectal” neoplasm in the title, abstract and manuscript given that rectal neoplasms have been included as well. (for example, “colonic and rectal lesions” could be revised as “colorectal lesions” in the abstract part.)

4. Please specify the “percentage” of patients with complications in the abstract part.

5. How to define “minor” perforation ?

6. Please use serrated sessile “lesions” rather than “polyps” in the manuscript and tables.

7. In “Table 2”, please specify the percentage of HGD/LGD lesions.

8. Please explain why using ESD for lesions less than 20mm. According to recommendations from international societies, colorectal LST which are less than 20mm could be managed by EMR.

9. Please define “long ESD time” in the Materials and Methods part rather than in the Result part.

10. Was submucosal fibrosis associated with long ESD time ?

11. What was the proficiency of endoscopists in EUS ? What kind the EUS used in this study (miniature probe with same frequency ? radial echoendoscope ?) ?

12. Please evaluate inter-observer and intra-observer variation for EUS interpretation.

13. Please send manuscript for English language editing.

6. PLOS authors have the option to publish the peer review history of their article (what does this mean?). If published, this will include your full peer review and any attached files.

Reviewer #1: No

Reviewer #2: No

---

## [Author Response · Author response to Decision Letter 0]

8 Sep 2022

For Editor Comments

1. As the study aimed to study ESD outcome in Southern Taiwan, please provide short discussion for their result with other reports of colonic resection from the same area with similar background of ESD development

Ans: We provided in Page 17, Line 221 to 228 

As for the result of colorectal ESD in the same area with a similar background, Choo et al. revealed en-bloc resection rate (72.7%) and R0 resection rate (66.7%) with perforation rate (15.2%) when the ESD technique was newly developed in Southern Taiwan [18]. A study by Tseng et al. showed en-bloc resection rate (90.2%) and R0 resection rate (89.1%) with perforation rate (12%) [19]. Although the studies mentioned above vary, improvements in en-bloc resection rate, R0 resection rate, and complication rate were observed in our study, contributing to the accumulation of ESD experience and improvement of ESD training programs in Southern Taiwanese hospitals.

For Reviewer #1:

1. The method of EUS (mini-probe or radial /linear scope).

Ans: We provided in Page 6, Line 98 to 99 

Our EUS procedures used a miniature Probe (UM-2R; Olympus Medical Systems, Tokyo, Japan) and an ultrasound system (EU-ME2 Premier Plus; Olympus Medical Systems, Tokyo, Japan). 

2. They should clearly mention that how many cases had endoscopic procedure before ESD (EMR or tried for polypectomy in another center)

Ans: We provided in Page 5, Line 74 to 75 

None of these patients received previous EMR or polypectomy. Sixteen cases underwent biopsy only, which were unrelated to submucosal fibrosis during ESD.

For Reviewer #2:

1. “Clinical outcomes of colon endoscopic submucosal dissection for colonic neoplasms: A single center experience in South Taiwan” => Revise as “Clinical outcomes of endoscopic submucosal dissection for colorectal neoplasms: A single center experience in Southern Taiwan.”

Ans: We already revised in Page 1, Line 1 to 3.

2. Please shorten the background and study aim in the abstract part.

Ans: Already shortened as Page 2, Row 21 to 24.

3. Please revise “colonic” neoplasm as “colorectal” neoplasm in the title, abstract and manuscript given that rectal neoplasms have been included as well. (For example, “colonic and rectal lesions” could be revised as “colorectal lesions” in the abstract part.)

Ans: Already revised and highlighted in file: Revised Manuscript with Track Changes

4. Please specify the “percentage” of patients with complications in the abstract part.

Ans: Already Specified in Page 2, Line 31 to 33.

5. How to define “minor” perforation?

Ans: Mentioned in Page 9, Line 140 to 142.

Our minimal perforation was defined as muscle layer defect without observation of mesenteric fat or intra-peritoneum organ and pneumoperitoneum.

6. Please use serrated sessile “lesions” rather than “polyps” in the manuscript and tables.

Ans: Already revised and highlighted in file: Revised Manuscript with Track Changes

7. In “Table 2”, please specify the percentage of HGD/LGD lesions.

Ans: Already revised and highlighted in file: Revised Manuscript with Track Changes

8. Please explain why using ESD for lesions less than 20mm. According to recommendations from international societies, colorectal LST which are less than 20mm could be managed by EMR.

Ans: Explained in Page 5, Line 78 to 81.

We chose ESD over EMR in some colorectal lateral spreading tumors less than 20 mm in cases of suspected lesions with limited submucosal invasion or difficult locations for en-bloc EMR, such as ileocecal valve, hepatic/splenic flexure, and sigmoid colon.

9. Please define “long ESD time” in the Materials and Methods part rather than in the Result part.

Ans: Revised in Page 7 to 8, Line 120 to 122.

10. Was submucosal fibrosis associated with long ESD time?

Ans: Mentioned in Page 19, Line 260-262

Although a study in Japan revealed fibrosis as a factor influencing long procedure times, there was no statistical significance in our study due to the limited cases of submucosal fibrosis.

11. What was the proficiency of endoscopists in EUS? What kind the EUS used in this study (miniature probe with same frequency? radial echoendoscope?)?

Ans: We provided in Page 6, Line 96 to 99

EUS procedures were performed by two experienced endoscopists who have performed more than 2000 EUS procedures. Our EUS procedures used a miniature Probe (UM-2R; Olympus Medical Systems, Tokyo, Japan) and an ultrasound system (EU-ME2 Premier Plus; Olympus Medical Systems, Tokyo, Japan). 

12. Please evaluate inter-observer and intra-observer variation for EUS interpretation.

Ans: We selected those six cases of discordance to evaluate inter-observer and intra-observer variation. 

 The Table was provided in file: Respond to Reviewers

13. Please send manuscript for English language editing.

Ans: Already done and the revision was highlighted in file: Revised Manuscript with Track Changes

---

## [Decision Letter · Decision Letter 1]

19 Sep 2022

PONE-D-22-23547R1Clinical outcomes of endoscopic submucosal dissection for colorectal neoplasms: A single-center experience in Southern TaiwanPLOS ONE

Dear Dr. Tai,

Thank you for submitting your manuscript to PLOS ONE. After careful consideration, we feel that it has merit but does not fully meet PLOS ONE’s publication criteria as it currently stands. Therefore, we invite you to submit a revised version of the manuscript that addresses the points raised during the review process. Some minor comments require further revision and we invite you to submit a manuscript revision according to reviewer's comments. 

We look forward to receiving your revised manuscript.

Kind regards,

Hsu-Heng Yen

Academic Editor

PLOS ONE

Journal Requirements:

Reviewers' comments:

Reviewer's Responses to Questions

**Comments to the Author**

1. If the authors have adequately addressed your comments raised in a previous round of review and you feel that this manuscript is now acceptable for publication, you may indicate that here to bypass the “Comments to the Author” section, enter your conflict of interest statement in the “Confidential to Editor” section, and submit your "Accept" recommendation.

Reviewer #1: All comments have been addressed

Reviewer #2: All comments have been addressed

2. Is the manuscript technically sound, and do the data support the conclusions?

Reviewer #1: Yes

Reviewer #2: Yes

3. Has the statistical analysis been performed appropriately and rigorously? 

Reviewer #1: Yes

Reviewer #2: Yes

4. Have the authors made all data underlying the findings in their manuscript fully available?

Reviewer #1: Yes

Reviewer #2: Yes

5. Is the manuscript presented in an intelligible fashion and written in standard English?

Reviewer #1: Yes

Reviewer #2: Yes

6. Review Comments to the Author

Reviewer #1: this revised form can be published . this is valuable as real life data. authors teplied all quesıns.

Reviewer #2: 1) Please use pathological findings as standard reference and calculate the interobserver and intraobserver variation which presented as kappa value.

2) Please provide a more clean version of manuscript with language editing parts deleted.

7. PLOS authors have the option to publish the peer review history of their article (what does this mean?). If published, this will include your full peer review and any attached files.

Reviewer #1: **Yes: **Filiz Akyuz

Reviewer #2: No

---

## [Author Response · Author response to Decision Letter 1]

20 Sep 2022

For Reviewer #2:

1) Please use pathological findings as standard reference and calculate the interobserver and intra-observer variation which presented as kappa value.

Ans:

We used SPSS to calculate Kappa value.

Intra-observer (Previous interpretation and interpretation in 2022): Kappa value: 1

Inter-observer (Previous interpretation and interpretation by another EUS endoscopist in 2022): Kappa value: 0.897

2) Please provide a cleaner version of manuscript with language editing parts deleted.

Ans: I had provided two versions of manuscript; one is clean version (File name: manuscript) and another one is track-change version (File name: Revised Manuscript with Track Changes).

---

## [Editor Report · Decision Letter 2]

23 Sep 2022

Clinical outcomes of endoscopic submucosal dissection for colorectal neoplasms: A single-center experience in Southern Taiwan

PONE-D-22-23547R2

Dear Dr. Tai,

We’re pleased to inform you that your manuscript has been judged scientifically suitable for publication and will be formally accepted for publication once it meets all outstanding technical requirements.

Kind regards,

Hsu-Heng Yen

Academic Editor

PLOS ONE